# VIDEO DEBLURRING WITH ADAPTIVE HIGH-FREQUENCY EXTRACTION

## ABSTRACT

State-of-the-art video deblurring methods use deep network architectures to recover sharpened video frames. Blurring especially degrades high-frequency information yet this aspect is often overlooked by recent models that focus more on enhancing architectural design. The recovery of high frequency detailing can be non-trivial, in part due to the spectral bias of neural networks. Neural networks are biased towards learning low frequency functions, making it to prioritize learning low frequency components. To enhance the learning of latent high frequencies, it is necessary to enforce explicit structures to capture the fine details or edges. This work merges the principles of the classic unsharp masking with a deep learning framework to emphasize the essential role of high-frequency information in deblurring. We generate an adaptive kernel, constructed from a convex combination of dynamic coefficients and predefined high-pass filtering kernels. This kernel is then employed in a spatio-temporal 3D convolution process to extract high-frequency components from the data. This method significantly improves video deblurring, achieving a noteworthy enhancement with an increase of up to 0.61dB in PSNR over top models on GORPO dataset. Additionally, it outpaces the majority of them in inference time.

## 1 INTRODUCTION

Video deblurring sharpens video frames from blurry input sequences. Deblurring is ill-posed as it aims to recover information lost during blurring. Classical sharpening techniques include unsharp masking and high-pass filtering to enhance the edges and detailing. Other methods (Xu et al. (2013); Krishnan et al. (2011); Pan et al. (2016a)) try to estimate the underlying blur kernels to reverse the effects. In the case of multi-image deblurring (Zhang et al. (2013); Cai et al. (2009)), the missing information can be approximated by observations over multiple frames. However, these classical methods are usually based on degradation models that are too simple to capture real-world scenes.

Recent advancements in video deblurring methods rely on deep neural networks trained end-to-end. These methods use advanced alignment techniques (Wang et al. (2022); Pan et al. (2020); Lin et al. (2022)) and deformable convolutions (Wang et al. (2019); Jiang et al. (2022)) which are effective but also computationally expensive and non-ideal for optimized hardware. Another hurdle confronting these models is the effective learning of high frequencies. Neural networks are prone to spectral bias - a preference for learning more towards low frequency component (Rahaman et al. (2019)). As a result, this inclination can hinder their ability to discern intricate textures or patterns. The bias against learning high frequencies poses a significant obstacle in video deblurring, where the recovery of these lost high-frequency elements is crucial. Recent works mitigate the spectral bias with the high-dimensional Fourier feature space (Tancik et al. (2020); Mildenhall et al. (2021)). Partially inspired by these efforts, our work utilizes simple high-pass filtering kernels, discovering they alone can yield substantial improvement.

Unsharp masking (Deng (2010); Ye & Ma (2018)) is a classical image sharpening technique which emphasizes high-frequency information by enhancing the image gradients. Inspired by the formulation of unsharp masking, we extend it to incorporate into a deep learning framework and customise it towards video deblurring. Central to unsharp masking is the explicit extraction of high-frequency components from the image. Integrating similar components (spatial and temporal gradients) into a neural network for video deblurring leads to a marked improvement (0.39dB), highlighting the

benefits of explicit high-pass filtering operations for deblurring. Inspired by these results, and to mitigate spectral bias, we propose to use a set of predefined high-pass filtering kernels for video deblurring. These kernels act as building blocks to learn high-frequency extraction kernels for our adaptive high-frequency extraction network (AHFNet). AHFNet attains state-of-the-art performance on video deblurring datasets while limiting computation complexity.

Our contributions are as follows:

- We reformulate video deblurring by extending unsharp masking. Explicitly extracting high-frequency information is critical and including spatial and temporal gradients greatly enhances video deblurring.

- We present a new high-frequency extraction operation, designed to adaptively extract high frequencies. The integration of operation into the network effectively captures high-frequency components, allowing neural networks to utilize the detailed information in the input video fully for the restoration of lost details.

- We have conducted both quantitative and qualitative evaluations of our proposed AHFNet on GOPRO and DVD datasets. AHFNet delivers state-of-the-art performance in terms of accuracy. Efficiency metrics (GMACs and runtime) further underline the model's efficacy. Notably, our model achieves a maximum speedup of 35x compared to other models while maintaining superior PSNR and SSIM scores.

## 2 RELATED WORK

**Classic deblurring algorithms.** Classic deblurring algorithms often leverage image structures or priors, e.g. sparse gradient priors (Xu et al. (2013); Krishnan et al. (2011)), intensity priors (Pan et al. (2016a)), and edges (Cho & Lee (2009); Xu & Jia (2010); Yang & Ji (2019); Sun et al. (2013); Xu et al. (2013)). Unsharp masking (Ye & Ma (2018); Deng (2010); Polesel et al. (2000)) counteracts blurring in an image by reintroducing a scaled version of the image's high-frequency components. This emphasizes the fine details or edges and enhances the image's overall sharpness. Other methods iteratively estimate the blur kernel and sharpened outputs (Cho & Lee (2009); Zhang et al. (2022b); Pan et al. (2016b); Shan et al. (2008)). Multi-image blind deconvolution algorithms restore the target image from multiple blurred or relevant images (Cai et al. (2009); Rav-Acha & Peleg (2005); Zhu et al. (2012); Zhang et al. (2013); He et al. (2012)).

Classic deblurring algorithms have difficulty in generalizing well to diverse and unseen types of blur. In our work, we extend unsharp masking to the realm of deep learning. By doing so, we combine the advantages of utilizing high-frequency information with the substantial capacity of neural networks to enhance video deblurring.

**Deep learning methods.** Deep learning methods for video deblurring restore sharp videos with deep neural networks (Su et al. (2017); Wang et al. (2019); Pan et al. (2020); Zhong et al. (2020)). Recent works have focused on improving the information retrieval from neighboring frames using optical flow (Wang et al. (2022); Lin et al. (2022)), deformable convolution (Wang et al. (2019); Jiang et al. (2022)), and feature matching (Ji & Yao (2022); Li et al. (2021)). For example, deformable convolution requires increased memory access, irregular data structuring, and dynamic computational graphs , all adding to the complexity and resource demands (Guan et al. (2022)).

Existing works differ primarily in their architectural approaches to deblurring. Our work takes a distinct direction as we place a greater emphasis on the significance of high frequencies to mitigate the effects of spectral bias (Rahaman et al. (2019)) and improve deblurring outcomes.

**Kernel prediction networks.** Kernel prediction networks determine convolution weights dynamically during inference (Chen et al. (2020); Ma et al. (2020); Zhou et al. (2019); Jia et al. (2016)). The kernel prediction network (Xia et al. (2020); Zhang et al. (2023)) synthesizes kernel bases and coefficients to derive an adaptive filter. Nonetheless, the optimization of kernel prediction modules presents a significant challenge, requiring extensive data and extended training time to accurately predict a blur kernel. In our proposed method, we incorporate specific high-pass filtering kernels. This not only emphasizes the extraction of high-frequency components but also eases the computational load by solely concentrating on the prediction of coefficients.

## 3 APPROACH

### 3.1 FORMULATION

Our formulation is inspired by the relationship between unsharp masking and the residual learning. In one specific formulation of unsharp masking, the restored image $\hat{y}$ is the sum of input $x$ (presumably blurry) and some high-frequency components of $x$ extracted by convolving with high-pass filter kernel $K$:

$$\hat{y} = x + \lambda(K * x), \tag{1}$$

where $*$ denotes a convolution operation and $\lambda$ is a scaling factor that controls the amount of high-frequency contents to be added back. $K$ can be any high-pass filter, e.g. a Laplacian. In more general forms of unsharp masking, $\lambda$ can also be spatially varying and depend on the local information contained in $x$, i.e. $\lambda(x)$ (Ye & Ma (2018)). Eq. 1 has the same form as residual learning in deep neural networks, where $\lambda(K * x)$ serves as a residual term.

Assume a latent sharp image $y$ is blurred with a low-pass kernel $\tilde{K}$, resulting in a blurred image $x = \tilde{K} * y$. The actual residual is $y - x = (1 - \tilde{K}) * y$. Here, $(1 - \tilde{K})$ acts as a high-pass filter. Hence, the residual $(1 - \tilde{K}) * y$ is effectively the high-pass filtered result, or high frequencies, of $y$.

Given the importance of high-frequency information for deblurring, we extend the formulation of unsharp masking to video deblurring for a deep neural network. Specifically, we generalize $K$ and $*$ as a spatio-temporal 3D kernel and the corresponding convolution operator. The $\lambda$ scaling can be viewed as an adaptive transformation operator based on the characteristic of $x$. This generalization turns the Eq 1 into:

$$\hat{y}_t = x_t + \mathcal{F}(x_t, \mathcal{M}(\{x_i\}_{i=t-l}^{t+l})), \tag{2}$$

where $t$ denotes the frame index in the video, $\mathcal{F}$ is general transformation replacing $\lambda$, $\mathcal{M}$ is the spatio-temporal 3D high-frequency extraction operation and $\{x_i\}_{i=t-l}^{t+l}$ denotes the local temporal window around $x_i$ with span of $l$. This generalization includes traditional unsharp masking as a special case. Specifically, when $\mathcal{M}$ acts as high-pass filtering and $\mathcal{F}$ operates as scalar multiplication independent of input $x$, Eq 2 reduces to Eq 1.

Computing high-frequency features from a local temporal window is essential because blurring often involves inter-frame effects. Consider the GOPRO dataset as an example. A blurry frame is formed by accumulating sampled sharp frames, expressed as:

$$x \approx g(\frac{1}{T}\sum_{i=1}^{T} s_i), \tag{3}$$

where $g$ is the nonlinear camera response function (CRF), $T$ is the number of sampled frames and $s_i$ is the original frame captured during the exposure time. If we ignore the CRF, Eq 3 can be viewed as the convolution of consecutive frames with an averaging kernel. Therefore, the neighboring frames are required in video unsharp masking. We show the effects of the spatio-temporal high-frequency extraction in Section 4.3.

Our formulation is more flexible and expressive in terms of both high-frequency extraction and scaling, which is shown to be important in traditional unsharp masking. In other words, we decompose the deblurring operation into two stages: 1) extracting the high-frequency components; 2) transforming the high frequencies into residual. Instead of blindly training a black box to restore the residual, we emphasize the importance of extracting HF information from the video unsharp masking view.

### 3.2 ADAPTIVE HIGH-FREQUENCY EXTRACTION OPERATION

A key challenge of the high-frequency extraction module $\mathcal{M}$ is the learning bias of neural networks (Rahaman et al. (2019); Arpit et al. (2017)). Neural networks prioritize learning simpler patterns, taking more time to adapt to complex textures or fine details. To address this, we propose our adaptive high-frequency extraction operation, as presented in Figure 1b. Before delving into the details, we establish a proposition as follows and give proof in Appendix A:

**Proposition 1.** *Given $M$ spatial high-pass filters with impulse responses $h_i(x)$ and corresponding frequency responses $H_i(f)$ with cutoff frequencies $f_{ci}$ (sorted such that $f_{c1} \leq f_{c2} \leq \cdots \leq f_{cM}$, a linear combination of these filters with positive coefficients $\alpha_i$ in the spatial domain, represented as:*

$$h(x) = \sum_{i=1}^{M} \alpha_i h_i(x),\tag{4}$$

*will itself act as a high-pass filter in the spatial domain, with a corresponding frequency response $H(f)$ and a cutoff frequency not greater than $f_{c1}$.*

This motivates us to train a kernel prediction module whose weights are the convex combination of the predetermined kernels and the coefficients that are dynamically predicted. The main difference from other kernel prediction modules is that we limit the space of the predicted kernels within high-pass filtering kernels only. We consider a space that is formed by a group of 3D linearly independent high-pass filter kernels $\{k_i\}_{i=1}^{M} = k_1, \ldots, k_M$. Each kernel is of size $R^{T_k \times H_k \times W_k}$ where $T_k$, $H_k$ and $W_k$ represent the temporal length, height and width of the kernel, respectively. To obtain the high-frequency feature for $x_t$, we use the coefficient generator $\mathcal{G}$ to compute a coefficient $\alpha_t \in R^M$:

$$\alpha_t = \mathcal{G}(\{x_i\}_{i=t-l}^{t+l}),\tag{5}$$

The concatenated input of $T_k$ consecutive frames has a size of $C \times T_k \times H \times W$, where $C$ is the input dimension. The synthesized kernel $k_t$ has a size of $1 \times T_k \times H_k \times W_k$. The synthesized kernel for the $t$-th frame is:

$$k_t = \sum_{j=1}^{M} \alpha_{tj} k_j \tag{6}$$

Proposition 1 assures that $k_t$ continues to be high-pass filtering kernel. We perform spatio-temporal 3D convolution on the image $x_t$ and $k_t$ and obtain the feature $r_t$.

To cover the orthogonal direction , we rotate the kernel bases by 90 degrees anticlockwise, forming a new set of kernel bases aligned in an orthogonal direction, which we denote as $\tilde{K} = \{\tilde{k}_i\}_{i=1}^{M}$. Next, we apply the coefficients $\alpha_t$ to $\tilde{K}$ as illustrated in equation 6, resulting in another feature, denoted as $\tilde{q}_t$. Finally, we amalgamate the data extracted from high frequencies in orthogonal directions by summing the absolute values of the feature maps $r_t$ and $\tilde{r}_t$. The output can be expressed as:

$$r_t = \{k_i * x_t, \tilde{k}_i * x_t, |k_i * x_t| + |\tilde{k}_i * x_t|\},\tag{7}$$

where $*$ represents 3D convolution. The incorporation of rotated kernels enhances the diversity of frequency extraction in varying directions. By introducing magnitude, we facilitate a more comprehensive aggregation of information from the two directions.

By utilizing a set of predefined high-pass building kernels, we are able to extract various features from the input images, all within a designated feature space designed specifically for high-frequency features. Their convex combination works still as a high-frequency extraction. This approach alleviates the optimization workload in the early module as it only requires predicting $M$ coefficients for each building kernel. In contrast, predicting entire kernels would necessitate the prediction of $T_k H_k W_k$ number of parameters. By adaptively predicting different coefficients for each frame, the model can adjust its frequency extraction based on input characteristics. We handle color input by applying the same kernel separately to each channel.

### 3.3 ARCHITECTURE

We present the overview of our entire model in Figure 1a. A total of $N + 1$ paths or extraction modules are used to extract the $N$ high-frequency features along with the original RGB feature. They use the same building kernels, but each path operates with distinct trainable parameters. Each path comprises preprocessing modules designated as $\mathcal{P} = \{\mathcal{P}^n\}_{n=0}^{N}$, paired with a feature extractor, labeled as $\mathcal{R} = \{\mathcal{R}^n\}_{n=0}^{N}$. $\mathcal{R}^0$ is designated as an identity function. The subsequent $N$ extraction

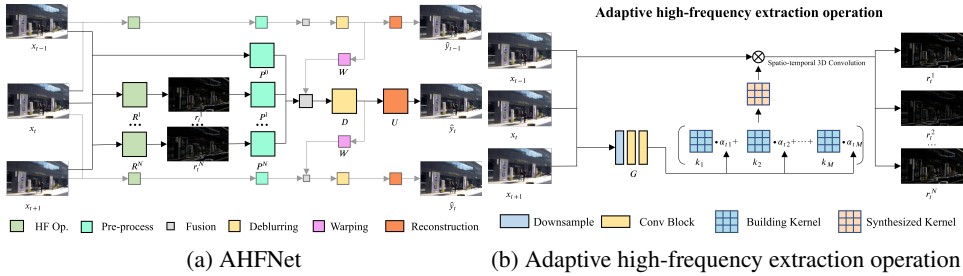

(a) AHFNet       (b) Adaptive high-frequency extraction operation

Figure 1: Overview of our model.

modules use the adaptive high-frequency extraction operation detailed in Section 3.2. Each extraction module takes the input sequence, $X$, and converts it into a unique type of high-frequency feature set, $\{r_t^n\}_{n=1}^N$ for the $t$-th frame. This output is then channeled through a respective preprocessing module, $\mathcal{P}^n$, to be combined with other features for further analysis. The result of the $n$-th path for the $t$-th frame is:

$$p_t^n = \mathcal{P}^n(\mathcal{R}^n(\{x_i\}_{i=t-m}^{t+m})). \tag{8}$$

The recurrent deblurring module $\mathcal{D}$ takes the feature set $\{p_t^n\}_{n=0}^N$ and the feature from the previous frame, denoted as $h_{\text{prev}}$, as input. We fuse them using concatenation. We experimented other fusion structures, but found that they are not critical to the performance. The warping module $\mathcal{W}$ warps $h_{\text{prev}}$ with the optical flow computed between $x_{\text{prev}}$ and $x$ for feature alignment (Chan et al. (2021)). As a result, the process of the deblurring module can be formulated as:

$$h_t = \mathcal{D}(\{p_t^n\}_{n=0}^N, \mathcal{W}(h_{\text{prev}}, x_{\text{prev}}, x_t)). \tag{9}$$

where $h_{\text{prev}}$ represents $h_{t-1}$ and $h_{t+1}$ during forward and backward pass, respectively. Our model is a bidirectional method. In Figure 1, we make the warping connection a bit transparent to highlight the main processing. Finally, the reconstruction module $\mathcal{U}$ takes the $h_t$ from the forward (superscript f) and backward pass (superscript b) as input to reconstruct the final output:

$$\hat{y}_t = x_t + \mathcal{U}(h_t^{\text{f}}, h_t^{\text{b}}) \tag{10}$$

The predicted residual $\mathcal{U}(h_t^{\text{f}}, h_t^{\text{b}})$ is equivalent to the term $\mathcal{F}(x_t, \mathcal{M}(\{x_i\}_{i=t-l}^{t+l}))$ in Eq 2.

# 4 EXPERIMENTS

## 4.1 EXPERIMENTAL SETTING

**Dataset.** We used the DVD dataset (Su et al. (2017)) and GOPRO dataset (Nah et al. (2017)) for evaluation. The DVD dataset comprises 61 training and 10 testing videos with 6708 blurry-sharp image pairs. The GOPRO dataset has 22 training and 11 testing sequences, with 3214 image pairs. Following (Pan et al. (2020)), we used the version without gamma correction. Each video is $1280 \times 720 \times 3$ in size.

**Evaluation metrics.** We evaluated performance with the peak signal-to-noise (PSNR) and SSIM (Wang et al. (2004)). The computational expense of each model was indicated with runtimes and number of giga multiply-accumulate operations (MAC) with respect to the frame size of $1280 \times 720 \times 3$. Lower runtimes and GMACs denote greater efficiency.

**Training Details.** For training, we used a Charbonnier loss (Charbonnier et al. (1994)) and the ADAM optimizer (Kingma & Ba (2014)) with default hyperparameters, i.e. $\beta_1 = 0.9$ and $\beta_2 = 0.999$. The initial learning rate was set to $2 \times 10^{-4}$ and decayed with a cosine restart strategy (Loshchilov & Hutter (2016)) with a minimum learning rate of $1 \times 10^{-7}$; the number of epochs between two warm starts was set to 300k. We trained our model for 600k iterations. For data augmentation, we applied random rotations and flipping. The batch size was set to 8, and the length of a training sequence was set to 10 frames. The size of the training patch was $256 \times 256$.

Table 1: Comparison with SOTA on GOPRO dataset.

| Model | PSNR | SSIM | GMACs | Params | Time(s) |
|---|---|---|---|---|---|
| EDVR (Wang et al. (2019)) | 26.83 | 0,8426 | 468.25 | 20.04 | 0.246 |
| DBN (Su et al. (2017)) | 28.55 | 0.8595 | 784.75 | 15.31 | 0.063 |
| IFIRNN ($c_2h_3$) (Nah et al. (2019)) | 29.80 | 0.8900 | 217.89 | 1.64 | 0.053 |
| SRN (Tao et al. (2018)) | 29.94 | 0.8953 | 1527.01 | 10.25 | 0.244 |
| ESTRNN ($C_{90}B_{10}$) (Zhong et al. (2020)) | 31.02 | 0.9109 | 215.26 | 2.47 | 0.124 |
| CDVD-TSP (Pan et al. (2020)) | 31.67 | 0.9279 | 5122.25 | 16.19 | 1.763 |
| MemDeblur (Ji & Yao (2022)) | 31.76 | 0.9230 | 344.49 | 6.99 | 0.152 |
| ERDN (Jiang et al. (2022)) | 32.48 | 0.9329 | 29944.57 | 45.68 | 5.094 |
| STDANet-Stack (Zhang et al. (2022a)) | 32.62 | 0.9375 | 6000.00 | 13.84 | 2.827 |
| MMP-RNN($A_9B_{10}C_{18}F_8$) (Wang et al. (2022)) | 32.64 | 0.9359 | 264.52 | 4.05 | 0.206 |
| AHFNet | 33.25 | 0.9439 | 461.35 | 6.75 | 0.144 |

**Architecture Details.** The preprocessing module $\mathcal{P}$ comprises of a one-layer convolution followed by two residual dense blocks that downsample the input by a scale factor of 0.25. The preprocessing module has a complexity of only $4.26\%$ of the GMACs compared with the deblurring backbone. The deblurring module $\mathcal{D}$ consists of a single convolution layer followed by 30 residual blocks without batch normalization (Lim et al. (2017)). The output of the residual blocks is upsampled using pixel shuffle modules (Shi et al. (2016)) to obtain the residual in $\mathcal{U}$. We use SPyNet (Ranjan & Black (2017)) as the warping module $\mathcal{W}$.

**Building kernels.** In our implementation, high-frequency extraction operation uses only $m = 4$ building kernels, with the first two being normalized Sobel and the other two being $[[0], [-1], [1]]$ and $[[1], [-1/2], [-1/2]]$. This configuration implies, for instance, that the first kernel will yield a filtered output represented as $0 \cdot x_{t-1} - x_t + x_{t+1}$. Each kernel has a size of $3 \times 3 \times 3$. We opt for these four kernels as bases owing to their simplicity and comprehensiveness. Other building kernels can also fulfill the same role as long as they function as linearly independent high-pass filtering kernels.

## 4.2 COMPARISON WITH THE STATE-OF-THE-ART

Tables 1 and 2 compare our model with the state-of-the-art models on the GOPRO and DVD datasets, respectively. For the GOPRO dataset, we employ metrics of distortion performance and efficiency for comparison. On the DVD dataset, we ensure a fair comparison by comparing our model with models of similar complexity.

Our model achieves the state-of-the-art performance with relatively fewer GMACs and runtime. Compared with MMP-RNN, our PSNR is 0.61dB higher, and runtime is only 70%, even though our GMACs is larger. Compared with ERDN and STDANet, our model is superior in

Table 2: Comparison with SOTA on DVD.

| Model | PSNR | SSIM |
|---|---|---|
| SRN | 30.53 | 0.8940 |
| IFIRNN ($c_2h_3$) | 30.80 | 0.8991 |
| EDVR | 31.82 | 0.9160 |
| CDVD-TSP | 32.13 | 0.9268 |
| ARVo (Li et al. (2021)) | 32.80 | 0.9352 |
| STDANet-Stack | 33.05 | 0.9374 |
| AHFNet | 33.19 | 0.9400 |

terms of both performance and efficiency. Overall, the GOPRO data features stronger blurring; this is shown indirectly since the same methods have higher PSNRs on the DVD than on GOPRO. Comparing results across the datasets, our model is better on the GOPRO dataset than on the DVD dataset, probably indicating that high frequency becomes more significant when the blurry artifacts are severe.

We provide visual comparisons with state-of-the-art models in Figure 2 and 3. Figure 2 shows that our model is superior in capturing the digits "55" compared to other models, and it produces the sharpest curve of the car. In Figure 3, we provide examples of the challenging recovery of tree details, which are difficult to achieve with high fidelity. While all models exhibit some degree of

blurriness, our model has fewer artifacts than others. For instance, STDAN displays a diagonal strip artifact that our model does not exhibit.

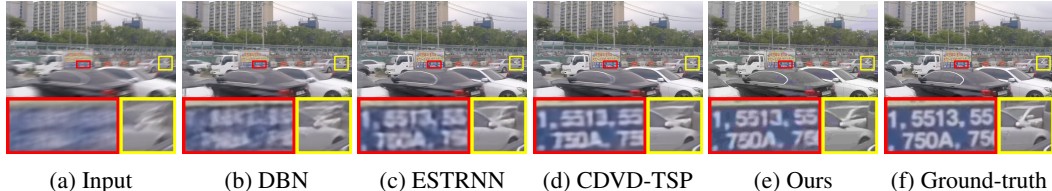

|        (a) Input        |        (b) DBN        |        (c) ESTRNN        |        (d) CDVD-TSP        |        (e) Ours        |        (f) Ground-truth        |

Figure 2: Qualitative comparisons on the GOPRO dataset.

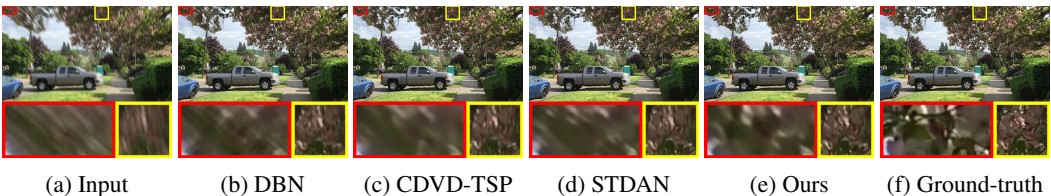

|        (a) Input        |        (b) DBN        |        (c) CDVD-TSP        |        (d) STDAN        |        (e) Ours        |        (f) Ground-truth        |

Figure 3: Qualitative comparisons on the DVD dataset.

### 4.3 EFFECTS OF HIGH-FREQUENCY EXTRACTION

In this section, we show the effects of introducing the high-frequency extraction in the formulation of Eq 2. In short, our findings reveal a marked enhancement in performance after integrating high-pass filters for video deblurring. Specifically, we incorporated several popular high-pass filtered kernels to an arbitrary baseline model. Our model followed the same architecture described in Section 3.3 but replaced the high-frequency extraction operation with simple high-pass filtering kernels. Since we use convolution-based kernel operation, we focused on the study on high-pass filters rather than adopting a Fourier transform perspective. Within the

Table 3: Experiments on various high-pass filters.

| Model | PSNR | SSIM | GMACs |
|---|---|---|---|
| RGB($\times$1) | 32.39 | 0.9346 | 387.25 |
| RGB$\times$2 | 32.47 | 0.9363 | 400.82 |
| RGB$\times$3 | 32.45 | 0.9356 | 414.39 |
| RGB+$\nabla x$ (Sobel) | 32.53 | 0.9363 | 403.76 |
| RGB+$\nabla x$ (Prewitt) | 32.54 | 0.9364 | 403.76 |
| RGB+$\nabla x$ (Kirsch) | 32.51 | 0.9356 | 403.76 |
| RGB+$\nabla^2 x$ | 32.37 | 0.9335 | 402.26 |
| RGB+$\nabla_t x$ | 32.41 | 0.9340 | 400.82 |
| RGB+$\nabla x + \nabla_t x$ | 32.78 | 0.9394 | 417.28 |

realm of high-pass filters, we used first-order filters ($\nabla x$), including Sobel, Prewitt, and Kirsch, second-order filters ($\nabla^2 x$) like the Laplacian, and the temporal gradient filter ($\nabla_t x$). This selection of simpler filters was intentional to restrain the growth in complexity. With a pre-processing module in place, these filtered results would still contribute to a highly intricate feature map. The temporal kernel used is $[[1/2], [-1], [1/2]]$.

Results are shown in Table 3. The variant "RGB$\times k$" repeated the early module ($\mathcal{P} \circ \mathcal{R}$) $k$ times so that the complexity matches the others for fair comparison. It can be observed that a modest increase in the complexity of the early modules enhances performance (from 32.39dB to 32.47dB), yet a further complexity increase reverses this gain, causing a decline to 32.45dB. This highlights that increasing complexity does not invariably improve performance.

The incorporation of the first-order filters enhances the PSNR across the various experimental variants, with improvement ranging from 0.04dB to 0.07dB. The second-order and temporal gradients fail to yield expected performance boosts. The former is more noise-sensitive, despite blur effects in the input theoretically suppressing noise, compared to the first-order gradient. The temporal gradient is affected by misalignment issues. In alignment with the discussion regarding the frequency calculation requirement of a temporal window in Section 3.1, merging the first-order and temporal filters results in a substantial improvement of 0.39dB. This approach generates a significant impact,

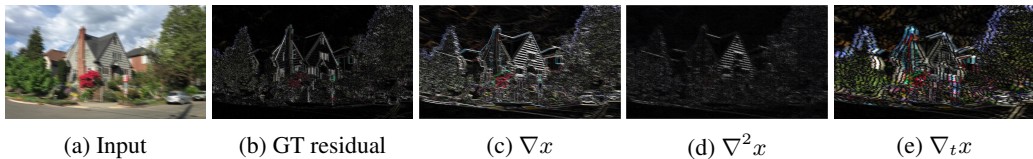

(a) Input       (b) GT residual       (c) $\nabla x$       (d) $\nabla^2 x$       (e) $\nabla_t x$

Figure 4: Visual examples for various image representations of the input.

Table 4: Comparison of the high-frequency extraction operation.

|  | w/o $\mathcal{R}$ | 3DConv | Naive Kernels | AHFNet | | | |
|---|---|---|---|---|---|---|---|
| $N$ | - | 6 | 6 | 0 | 2 | 4 | 6 |
| PSNR | 32.39 | 32.82 | 32.80 | 32.56 | 32.68 | 32.71 | 32.89 |
| SSIM | 0.9346 | 0.9384 | 0.9378 | 0.9366 | 0.9372 | 0.9371 | 0.9398 |
| GMACs | 387.25 | 462.71 | 461.35 | 410.73 | 427.66 | 444.51 | 461.35 |

boosting the PSNR by 0.39dB and SSIM by 0.0048 in comparison to the baseline model, and enhancing the PSNR by 0.33dB and SSIM by 0.0038 relative to a model of similar complexity. This improvement surpasses the aggregate benefits of employing them separately, showing the potential of temporal gradients to fine-tune high-frequency extraction from spatial gradients. The choice of building kernels of AHFNet is also motivated by the first-order gradient filters and temporal kernels.

We visualize various image representations derived from different high-pass filtering kernels in Figure 4. Visually, the first-order gradient map adeptly captures more details compared to the second-order gradient and proves more robust against the distortion apparent in the results of the temporal gradient. However, the straight lines in the triangular roof that are captured by the spatial gradient maps are not visible in the ground-truth residual map. This means that the spatial gradients contain unimportant details. On the other hand, such straight lines are invisible in the temporal gradient, indicating that it filters out of those sharp edges that do not receive significant blur.

### 4.4 HIGH FREQUENCY KERNEL OPERATION

We assess the effectiveness of the high-frequency extraction operation by comparing it with three variants: (A) (w/o $\mathcal{R}$) without $\mathcal{M}$ in Eq 2; (B) (3D Conv) using learnable 3D convolutions; and (C) (Naive Kernels) substituting building kernels with $T_k H_k W_k$ number of kernels, each with a unique non-zero entry. Each model is trained for 300k iterations. We compare with the latter two variants because they are as complex as ours but not specially designed to extract high frequencies. Additionally, we assess scenarios with varying numbers of paths for high-frequency extraction, i.e. $N$. The variant without $\mathcal{R}$ exhibits differing complexity compared to AHFNet($N = 0$) because of adjustments made in the pre-processing module to enhance representation. Table 4 shows that our proposed operation attains the highest PSNR and SSIM. Increasing the number of paths can further improve the performance. However, as illustrated in Section 4.3, a mere increase in complexity does not guarantee better results, especially when high-frequency extraction is not incorporated.

To evaluate the learning on different frequencies, we divide the Fourier coefficients into 10 frequency sub-bands based on their frequencies and calculate the average MSE on each sub-band on the GOPRO dataset. Specifically, consider an output image of size $H \times W$. The discrete Fourier transform calculates the frequencies at the coordinate $(i, j)$, where $i$ and $j$ range from $[-\frac{H}{2}, \frac{H}{2} - 1]$ and $[-\frac{W}{2}, \frac{W}{2} - 1]$, respectively. We define the length of each sub-band as $d = ((\frac{H}{2})^2 + (\frac{W}{2})^2)/10$. For the $z$-th frequency sub-band, we select the Fourier coefficients where $zd \leq \sqrt{(i - \frac{H}{2})^2 + (j - \frac{W}{2})^2} < (z + 1)d$. Each coefficient in the sub-band is then used to compute the MSE, which is compared with the corresponding coefficients in the Fourier transform. To make the difference obvious, we subtract the MSE values from those from the variant "w/o $\mathcal{R}$". The results are shown in Figure 6. Even with slight performance reductions in the highest frequency areas, possibly attributed to training variance, high-frequency extraction operation consistently excels in the majority of the sub-bands.

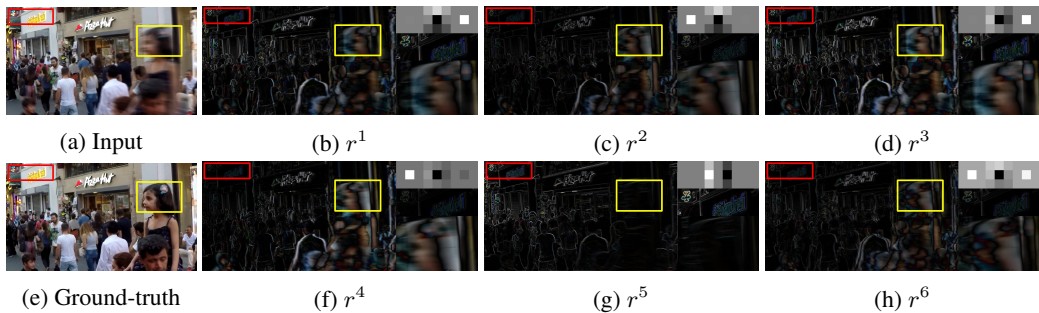

| (a) Input | (b) $r^1$ | (c) $r^2$ | (d) $r^3$ |

| (e) Ground-truth | (f) $r^4$ | (g) $r^5$ | (h) $r^6$ |

Figure 5: Examples of learned kernels and feature.

## 4.5 VISUALIZATION

We provide visualizations of the learned kernels and features, namely $\{k^n\}_{n=1}^N$ and $\{r^n\}_{n=1}^N$, as shown in Figure 5, for AHFNet. The figure displays six images (Figures 5b-5d and Figures 5f-5h), each corresponding to a different path. The synthesized six kernels are displayed in the top-right corner of the figure. Each kernel has dimensions $3 \times 3 \times 3$, i.e. $T_k = H_k = W_k = 3$. We use nearest-neighbor interpolation to scale the kernels up for better visualization. The top right part of the figure shows the generated kernels, and from left to right, the figure depicts the kernels that work on the previous, current, and next frames, respectively. Kernels are normalized to $[-1, 1]$ for visualization, where white, gray, and black signify 1, 0, and -1, respectively.

Our model's capability to extract diverse high-frequency information is observable, with no two synthesized kernel pairs being identical. All synthesized kernels emphasize the current frame, evident from the slightly gray color of all middle kernels which operates on the current frame. The fifth kernel exclusively examines the current frame, resulting in a spatial gradient map. The first and third kernels, differing in spatial domain orientations, underline our extended version's adaptability to various useful orientations for restoration.

## 5 CONCLUSION

We expand the unsharp masking algorithm for video deblurring, tapping into the advantage of utilizing high frequencies in unsharp masking coupled with the robust expressive capacity of deep neural networks. Experimental results underscore the significance of manual high-frequency extraction. Notably, the combination of first-order and temporal gradients substantially enhances performance.

Progressing further, we introduce adaptability in high-frequency extraction by generating the coefficients of several high-pass filter kernels. Initially, we show that using a linear combination with positive coefficients and high-pass filter kernels continues to function as a high-pass filter. This assurance confirms our generated kernel's capacity to efficiently extract high fre-

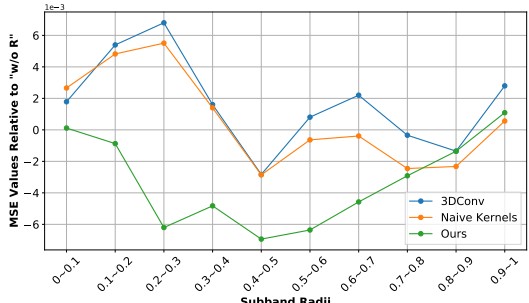

Figure 6: Comparison of learning in various frequency sub-bands. Lower MSE values indicate better performance.

quencies from the input image. By integrating this operation into our model, we attain the state-of-the-art results in video deblurring datasets in terms of both performance and complexity. Visual demonstrations further highlight the effectiveness of our proposed approach. Our approach shows enhanced performance across various frequency sub-bands and improved filtered results in the spatial domain.

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

## A HIGH-PASS FILTER KERNEL

The proof for Proposition 1 is presented below.

*Proof.* Consider $h(x) = \sum_{i=1}^{M} \alpha_i h_i(x)$. By utilizing the linearity of the Fourier transform, it can be expressed in the Fourier domain as:

$$H(f) = \sum_{i=1}^{M} \alpha_i H_i(f) \tag{11}$$

In this domain, $H_i(f)$, representing a high-pass filtering function, is generally observed as a non-decreasing function $g_i(f)$ ranging from 0 to 1. A value of $g_i(f) = 0$ signifies complete attenuation of the frequency component $f$, while $g_i(f) = 1$ denotes no attenuation.

Given that the sum and scalar multiplication of non-decreasing functions remain non-decreasing, $H(f)$ is also non-decreasing. Therefore, $h(x)$ is identified as a high-pass filter. $\square$

## B THE KERNELS FOR HIGH-PASS FILTERING

We describe the kernels we used in Section 4.3. The Sobel filter is defined as follows:

$$\text{Sobel}_x = \begin{bmatrix} 1 & 0 & -1 \\ 2 & 0 & -2 \\ 1 & 0 & -1, \end{bmatrix}, \text{Sobel}_y = \begin{bmatrix} 1 & 2 & 1 \\ 0 & 0 & 0 \\ -1 & -2 & -1 \end{bmatrix}. \tag{12}$$

The Prewitt filter is:

$$\text{Prewitt}_x = \begin{bmatrix} -1 & 0 & 1 \\ -1 & 0 & 1 \\ -1 & 0 & 1, \end{bmatrix}, \text{Prewitt}_y = \begin{bmatrix} -1 & -1 & -1 \\ 0 & 0 & 0 \\ 1 & 1 & 1 \end{bmatrix}. \tag{13}$$

The Kirsch filter is:

$$\text{Kirsch}_x = \begin{bmatrix} -3 & -3 & 5 \\ -3 & 0 & 5 \\ -3 & -3 & 5, \end{bmatrix}, \text{Kirsch}_y = \begin{bmatrix} 5 & 5 & 5 \\ -3 & 0 & -3 \\ -3 & -3 & -3 \end{bmatrix}. \tag{14}$$

The Laplacian filter is:

$$\text{Laplacian} = \begin{bmatrix} 0 & -1 & 0 \\ -1 & 4 & -1 \\ 0 & -1 & 0, \end{bmatrix} \tag{15}$$

## C ARCHITECTURE

The detailed architecture of our model is presented in Tables 5 to 8. For the pre-process module, Residual Dense Blocks (RDB) (Zhong et al. (2020)) are utilized as the foundational building blocks. The architectures of $\mathcal{P}^0$ and $\mathcal{P}^n$, where $n = 1, \ldots, N$, differ. Greater computational resources are allocated to $P^0$ due to its role in the final restoration in RGB format, prioritizing the direct RGB input over other features.

| Layer | Output | Coefficient Generator |
|---|---|---|
| conv1 | $T \times H/4 \times W/4 \times 3$ | $3 \times 3 \times 3$, stride 1 |
| conv2 | $T \times H/4 \times W/4 \times 3$ | $3 \times 3 \times 3$, stride 1 |
| AvgPool | $1 \times 1 \times 1 \times 3$ | - |
| Linear | $1 \times 1 \times 1 \times 3$ | $3 \times NM$ |

Table 5: Coefficient generator $\mathcal{G}$ architecture.

## D EXPERIMENTS

### D.1 WEIGHTS IN THE FUSION MODULE

| Layer | Output | Pre-process Module |
|-------|--------|--------------------|
| Conv1 | $H \times W \times 3$ | $5 \times 5$, stride 1 |
| RDB1 | $H \times W \times 3$ | $[3 \times 3, 16, \text{dense conv}] \times 4$ |
| Conv2 | $H/2 \times W/2 \times 32$ | $5 \times 5$, stride 2 |
| RDB2 | $H/2 \times W/2 \times 32$ | $[3 \times 3, 24, \text{dense conv}] \times 4$ |
| Conv3 | $H/4 \times W/4 \times 64$ | $5 \times 5$, stride 2 |

Table 6: Pre-process module $\mathcal{P}^0$ architecture.

| Layer | Output | Pre-process Module |
|-------|--------|--------------------|
| Conv1 | $H \times W \times 3$ | $5 \times 5$, stride 1 |
| RDB1 | $H \times W \times 3$ | $[3 \times 3, 16, \text{dense conv}] \times 2$ |
| Conv2 | $H/2 \times W/2 \times 16$ | $5 \times 5$, stride 2 |
| RDB2 | $H/2 \times W/2 \times 16$ | $[3 \times 3, 24, \text{dense conv}] \times 2$ |
| Conv3 | $H/4 \times W/4 \times 16$ | $5 \times 5$, stride 2 |

Table 7: Pre-process module $\mathcal{P}^n, n = 1, \ldots, N$ architecture.

| Layer | Output | Deblurring Module |
|-------|--------|-------------------|
| Conv1 | $H/4 \times W/4 \times 64$ | $3 \times 3$, stride 1 |
| ResBlock1 | $H/4 \times W/4 \times 64$ | $\begin{matrix} 3 \times 3, 64 \\ 3 \times 3, 64 \end{matrix} \times 30$ |
| Transposed conv1 | $H/2 \times W/2 \times 32$ | $3 \times 3$, stride 2 |
| Transposed conv2 | $H \times W \times 16$ | $3 \times 3$, stride 2 |
| Conv1 | $H \times W \times 3$ | $5 \times 5$, stride 1 |

Table 8: Deblurring module $\mathcal{D}$ architecture.

Figure 7 presents a visualization of the weight of the first convolution in $\mathcal{D}$, which takes the concatenation of the inputs in Eq.9. The model fuses 224 channels, with the first 64 channels representing the RGB input, followed by $6 \times 16$ channels for 6 different features, and finally 64 channels for the warped output from the previous frame. The histogram values are scaled with respect to the input scale to ensure that the comparison is meaningful. The channel for $p^5$ represents the spatial gradient (see Figure 5g). The fusion convolution assigns more weight to the input, the spatial gradient map, and the previous warped results, which further illustrates the importance of the spatial gradient in deblurring.

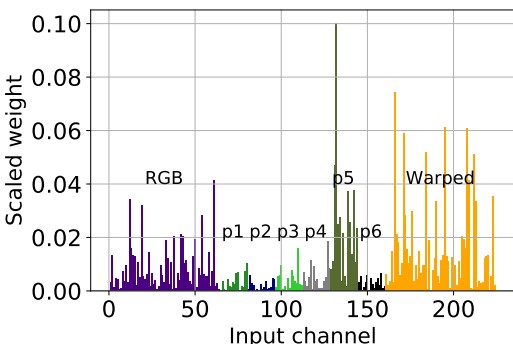

Figure 7: Histogram of the weights in the fusion.

## D.2 VISUAL COMPARISON

Additional visual comparisons are provided in Figures 8 to 11.

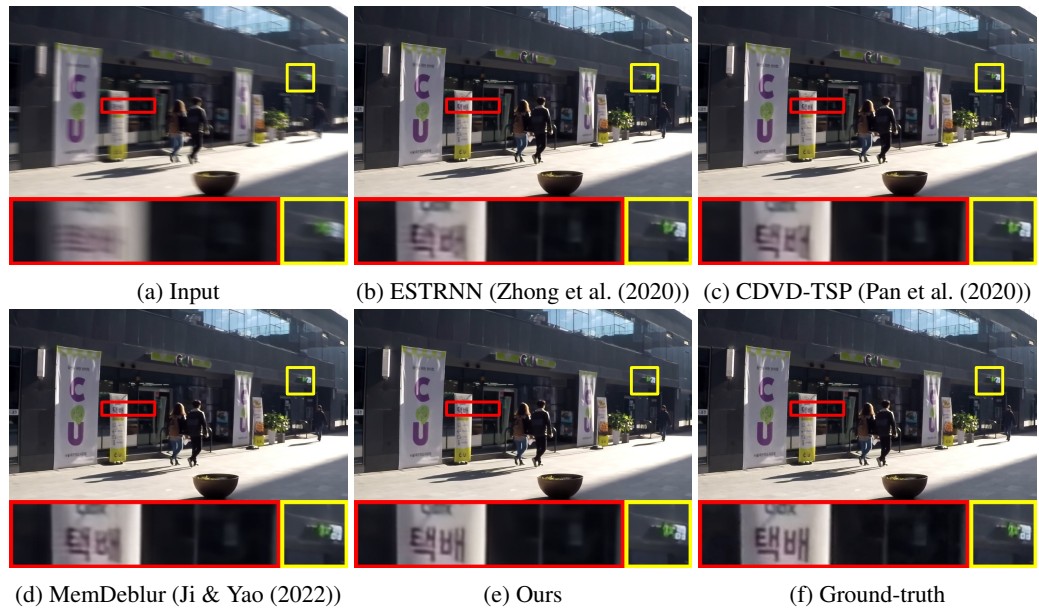

Figure 8: Qualitative comparisons on the GOPRO dataset.

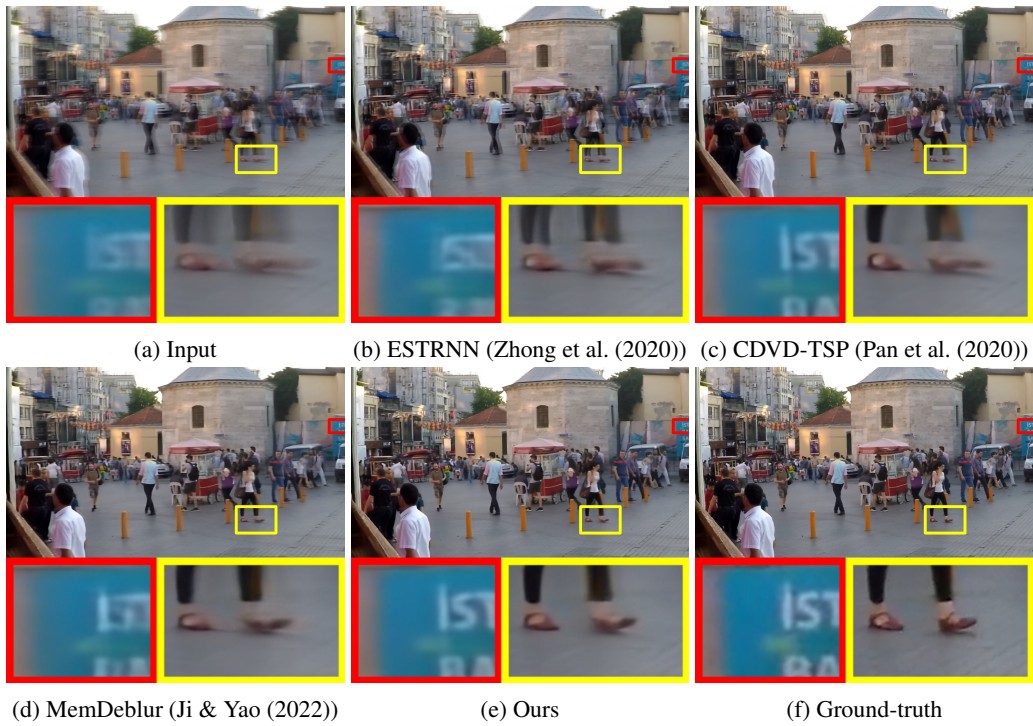

Figure 9: Qualitative comparisons on the GOPRO dataset.

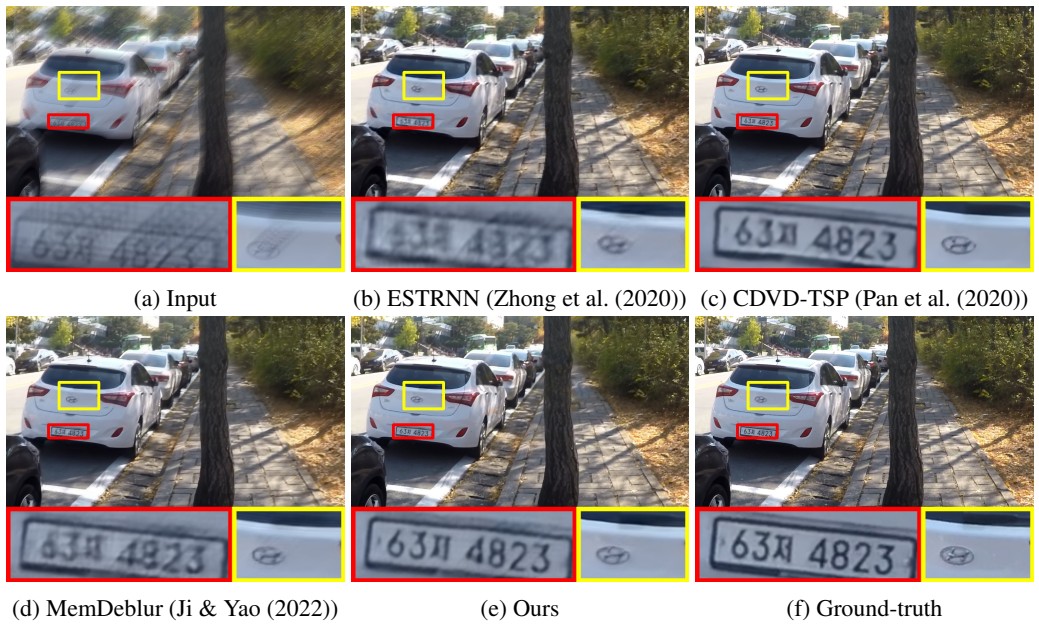

Figure 10: Qualitative comparisons on the GOPRO dataset.

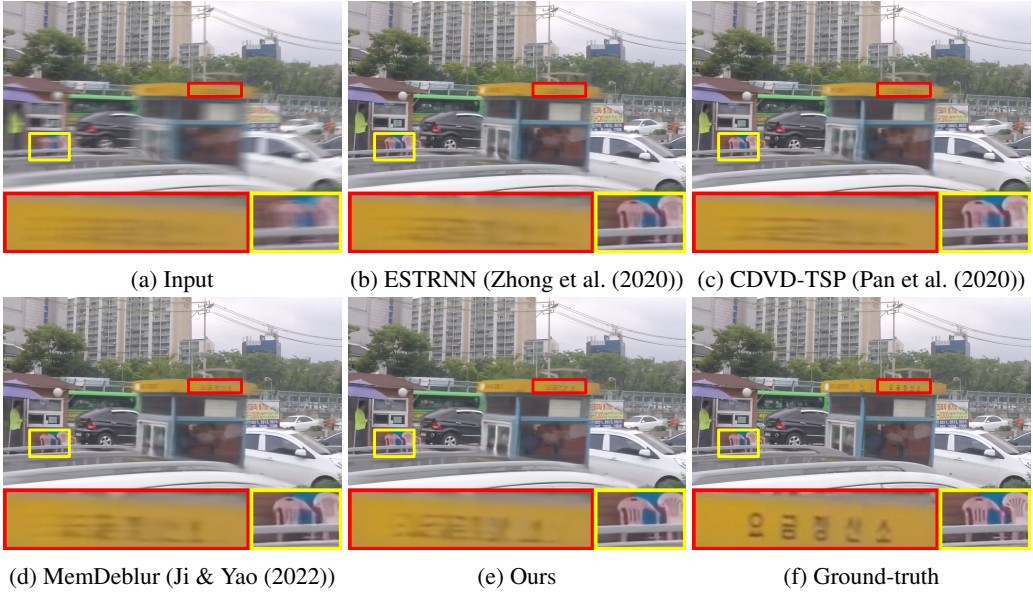

Figure 11: Qualitative comparisons on the GOPRO dataset.

