# OpenReview forum: "Video Deblurring with Adaptive High-frequency Extraction"
_ICLR.cc/2024/Conference — Submitted to ICLR 2024_

### Official Review · Reviewer_Ebb6 · 2023-10-13

**Soundness:** 2 fair
**Presentation:** 1 poor
**Contribution:** 1 poor
**Rating:** 3
**Confidence:** 5

**Summary:**

This paper proposes a network for dynamic scene image deblurring by adaptively fusing extracted high-frequency information. The provided experiments show the proposed method performs better than existing methods.

**Strengths:**

The provided results show the proposed method performs better than other methods (even though some improvement is marginal).

**Weaknesses:**

1. The motivation that unsharp masking is useful for deblurring is not clear. The only thing used in the proposed network is extracting high-frequency information.
2. The only novelty of this paper is adaptive extracting high-frequency information which is too marginal.
3. Why does the network use pre-defined high-frequency kernels to extract information? Why not allow the network to learn it directly?
4. The authors do not provide enough analysis to demonstrate the effectiveness of the high-frequency information. More analysis is needed.
5. Some improvement is too marginal, e.g. Table 2 and Figure 2(with artifacts).

**Questions:**

See weaknesses for details.

---

> ### Author Response · Authors · 2023-11-22
>
> Thank you for noting the effectiveness of our proposed method and questions regarding our methodology.
>
> *Q1: The motivation is not clear.*
>
> **A1:** The primary goal of video deblurring is to recover high-frequency (HF) details that are typically lost in the blurring process. This recovery of HF information is essential. We employ the unsharp masking technique, a strategy that specifically targets HF components, to aid in retrieving these lost details. This approach underscores the importance of HF information in the deblurring process.
>
> *Q2: The novelty of adaptive extracting HF information is marginal.*
>
> **A2:** Our approach introduces a novel perspective in video deblurring, drawing inspiration from the classic technique of unsharp masking. This unique focus on HF information sets our work apart, as no other studies have explored deblurring through this specific lens, despite its obvious relevance. Our method for extracting HF details goes beyond black box learning with neural networks, which are typically unable to discern such nuances on their own.
>
> *Q3: Why does the network use pre-defined high-frequency kernels to extract information? Why not allow the network to learn it directly?*
>
> **A3:** We utilize pre-defined high-frequency kernels because learning a kernel directly without explicit specifications does not guarantee the creation of high-pass filters. The pre-defined high-frequency kernels allow the predicted kernel to be exactly a high-pass filter and eases the computational load by solely concentrating on the prediction of coefficients.
>
> Our experiments, shown in Table 4, compare direct kernel learning to variations of 3D convolution or "Naive Kernels" (when using a linear combination of directly learned kernels). We observed a performance decrease of 0.07dB and 0.09dB in these methods.
>
> *Q4: More analysis on the effectiveness of the HF information.*
>
> **A4:** We have conducted extensive experiments to demonstrate our method's effectiveness, including distortion performance analysis (Table 1-3), visual comparisons (Figure 2-3), and ablation studies on high-frequency extraction techniques (Table 4). Additionally, we provide visualizations of the representations (Figure 4), the generated kernels (Figure 5), and their performance across various frequency sub-bands (Figure 6).  Taken collectively, we think this presents a convincing case for the effectiveness of our approach on the HF information.
>
> We are open to specific suggestions for additional analysis.  Alternatively, we welcome specific questions about the effectiveness which can guide us in providing more analysis.
>
> *Q5: Some improvement is too marginal, e.g. Table 2 and Figure 2(with artifacts).*
>
> **A5:** In Table 2, our model outperforms STDANet-Stack by 0.14dB, a significant achievement given that STDANet-Stack is over 10 times more complex, as shown by their GMACs in Table 1. The disparity in complexity across various video deblurring models makes direct comparisons challenging. For instance, the complexity range in the models compared in Table 1 spans a factor of 139. A more balanced assessment is presented in our ablation study (Table 3), where our method boosts performance by 0.39dB with only a 7.7% increase in GMACs. This underlines the efficiency of our HF extraction technique in deep models.
>
> Regarding Figure 2, some artifacts are present due to the severe blurring of the input. However, the key improvement is our model's ability to differentiate details, such as clearly distinguishing the "55" at the top row, where models like CDVD-TSP fail.

---

### Official Review · Reviewer_5xHX · 2023-11-01

**Soundness:** 3 good
**Presentation:** 3 good
**Contribution:** 3 good
**Rating:** 8
**Confidence:** 2

**Summary:**

The paper proposes a type of kernel prediction network where the basis function are composed of high pass filters for video deblurring.

**Strengths:**

The paper proposes to use high pass filters as basis for kernel prediction. This is a very intuitive idea, and this makes the number of parameters to be learned smaller. The proposed method is able to perform better than the SOTA.

**Weaknesses:**

While the intuition for the development of the method is given, certain aspects of the results are not well-explained. For eg., based on the design one may expect the results from the proposed method to be sharper than the SOTA. But, what we are seeing is more than that. In Fig. 2 and Fig. 3, we see more details in the proposed method than in the compared methods. What is causing this to happen?

I would be interesting to how the method performs if the basis functions were not used, but instead KPN is used like it is generally without any restrictions. Such an experiment would help determine the usefulness of using the high pass filters as basis.

The proposed method is not video-specific. The idea and solution both are more single image/ burst imaging specific. The authors are applying the idea in video deblurring domain, but the proposed method is not specific to this. Is there a reason this method applied specifically for video deblurring?

**Questions:**

Check the weakness section.

---

> ### Author Response · Authors · 2023-11-22
>
> Thank you for recognizing the intuitiveness and efficiency of our proposed method, as well as for your insightful observations on the results.
>
> *Q1: The unexpected detailed results compared to SOTA.*
>
> **A1:** The sharpness typically relates to the clarity of edges and fine structures in an image. Our method's focus on HF extraction aids in identifying and recovering subtle textures and intricate patterns that are often lost or overlooked in standard deblurring processes. This not only enhances sharpness but also reconstructs finer details, contributing to the overall richness and depth of the image.
>
> *Q2: KRN-like experiment to determine the usefulness of basis functions.*
>
> **A2:** In Table 4, we demonstrate what happens when basis functions are not used. The method is either "3D conv" or "Naive kernels" variants (when using the linear combination).
> The "Naive kernels" variant closely represents the KPN method you referenced, showing a performance decrease of 0.09dB.
>
> *Q3: The method is not video-specific.*
>
> **A3:** Your question highlights an important aspect of our method. We intentionally designed it to be versatile, suitable for both single image and video deblurring cases.
>
> We chose to focus on video deblurring due to its complexity and the unique challenges it presents, such as temporal gradients and motion consistency across frames. This focus allows us to demonstrate our method's robustness and effectiveness in a more dynamic and challenging environment, highlighting its versatility and potential in various deblurring scenarios.

---

### Official Review · Reviewer_a8DG · 2023-11-01

**Soundness:** 2 fair
**Presentation:** 3 good
**Contribution:** 2 fair
**Rating:** 5
**Confidence:** 4

**Summary:**

The authors propose to extend unsharp masking using deep neural networks for video deblurring. The solution incorporates spatio-temporal 3D convolutions and high-pass filtering to focus on higher frequency features. Experiments on GoPro and DVD datasets show quantitatively and qualitatively improved results compared to previous works. Further experiments and analysis study the effectiveness of extracting high-frequency features and the proposed adaptive high-frequency operation.

**Strengths:**

Originality: The attempt to formulate unsharp masking using a deep neural network and its application to video deblurring is seemingly original.

Clarity: The paper is well-written, and the construction of the formulation for the solution is easy to follow by reading the paper.

Significance: The results look significant compared to the considered previous works.

**Weaknesses:**

My understanding is that the experimental results are presented for a single run per experiment. To better understand the significance of the claimed improvements, it is necessary to run the main experiments multiple times with a set of random number generator seeds and to present the results with properties of their distributions, such as mean and standard deviation under the assumption of a normal distribution.

**Questions:**

I am interested to see the performance of the proposed method compared to previous works on a real-world deblurring dataset, such as [1].

[1] Zhong, Zhihang, et al. "Real-world video deblurring: A benchmark dataset and an efficient recurrent neural network." *International Journal of Computer Vision* 131.1 (2023): 284-301.

---

> ### Author Response · Authors · 2023-11-22
>
> Thank you for appreciating the originality and clarity of our paper, and for your valuable suggestion on the experimental results.
>
> *Q1: Run experiments multiple times to present the result.*
>
> **A1:** This is a solid suggestion. We will add the mean and the standard to the reported table in the revision.
>
> *Q2: Performance on the real-world deblurring datasets.*
>
> **A2:** Transferring models to real-world datasets requires careful consideration. We will add the comparison on this real-world dataset [1] in the revision!
>
> [1] Zhong, Zhihang, et al. "Real-world video deblurring: A benchmark dataset and an efficient recurrent neural network." International Journal of Computer Vision 131.1 (2023): 284-301.

---

> > ### Comment · Reviewer_a8DG · 2023-12-05
> > **Revised score**
> >
> > After reviewing all the comments by other reviewers and the responses by the authors, I am lowering the score to help the average score converge towards a decision.
> >
> > Reasons:
> >
> > 1) Authors refer to a revision multiple times in their response. However, I could not find the requested material in the latest version of the paper. Missing promises include providing average and standard deviation of quantitative results and comparing on real world blur datasets.
> >
> > 2) Looking again at the complaints by the reviewers and especially on results shown in Table 4, it seems like the improvements given by the proposed AHFNet are marginal as compared to naive kernels (+0.09 for PSNR and +0.002 for SSIM), while the GMACs measure is also the same.
> >
> > 3) Responses to requests for more comparisons to the SOTA by reviewer HuzH were not convincing.

---

### Official Review · Reviewer_HuzH · 2023-11-02

**Soundness:** 2 fair
**Presentation:** 2 fair
**Contribution:** 2 fair
**Rating:** 3
**Confidence:** 5

**Summary:**

This paper proposes to merge the principles of the classic unsharp masking with a deep learning framework to emphasize the role of high-frequency information in deblurring. Specifically, it constructs the kernel with a convex combination of estimated coefficients and predefined high-pass filtering kernels. Several experiments are conducted to demonstrate its effectiveness.

**Strengths:**

High-frequency information are explicitly extracted to enforce the capture of finer details.

**Weaknesses:**

There are several unclear statements listed as follows.

**Questions:**

1. This work is not well motivated. This paper is proposed based on a statement that neural networks prefer learning low-frequency components. While the work [a] has analyzed that convs are high-pass filters. Please discuss in detail whether these two statements conflict.
[a] Park and Kim, How do vision transformers work? ICLR 2022.
2. In the third paragraph of Section 1, the authors state that integrating spatial and temporal gradients into a neural network for video deblurring leads to an improvement of 0.39dB. Where does the result come from?
3. The authors mention several times that they utilize a set of predefined high-pass building kernels. How are these kernels predefined?  Are they the simple kernels defined in “Building kernels”? They are too simple to extract sufficient high-frequency information. What about learning these kernels?
4. In the experiments, please add the comparisons with more recent methods, e.g.,
[b] Li et al., A Simple Baseline for Video Restoration with Grouped Spatial-temporal Shift, CVPR 2023.
[c] Liang et al., Recurrent Video Restoration Transformer with Guided Deformable Attention, NeurIPS 2022.
5. Compared to the recent video deblurring methods, e.g., [b] and [c], the proposed method does not achieve the state-of-the-art performance, i.e., at least 1.97dB worse than [b] on the dataset ofGoPro and 1.11dB worse than [c] on the dataset of DVD.
6. Please add more comparisons on real examples.

---

> ### Author Response · Authors · 2023-11-22
>
> Thank you for acknowledging our approach in extracting high-frequency information, and for your detailed questions and observations.
>
> *Q1: Conflict about neural networks' preference for low versus high-frequency components.*
>
> **A1:** Thank you for your intriguing question! We are not sure about the contradiction and a direct comparison is challenging because the ICLR’22 work does not discuss spectral bias. The experimentation in the two works are also not aligned. An in-depth comparison that could clarify this discrepancy would be intriguing, but it well-exceeds the scope of our current research on video deblurring.
>
> We note however, that spectral bias in neural networks [1] is well-established and cited by hundreds of papers. This idea forms the foundation of our study. Our experimental results confirm that integrating the high-frequency extraction operation enhances the information extraction in both low and high frequency sub-bands in Figure 6 and results in an remarkable improved performance in Table 3 and 4.  In that regard, regardless of the contradiction, we find the basis of spectral bias sufficient for our work.
>
> [1] N. Rahaman et al., On the spectral bias of neural networks, ICML 2019
>
> *Q2: Where does the result of the improvement of 0.39dB come from?*
>
> **A2:** 0.39dB comes from Table 2 by subtracting the “RGB" (32.39dB) from “RGB+$\nabla x$ + $\nabla_t x$" (32.78dB).
>
> *Q3: How are the high-pass building kernels predefined? Are the pre-defined kernels defined in "Building kernels"? What is the effect of learning these kernels?*
>
> **A3:** We define high-pass filter kernels by fixing their weights as the standard high-pass filters, such as derivative filters. The pre-defined filters are proven to be effective in Table 3. The actual high-pass filter is further predicted adaptively from the pre-defined kernels.
>
> These predefined kernels are not utilized in all building blocks of the network. Instead, we employ them in the network's early stages, where they prove to be adequately effective.
>
> Learning an arbitrary kernel directly does not guarantee that the resulting kernels will be high-pass filters; they could instead function as low-pass filters. To mitigate this, predefined kernels are employed as constraints, ensuring that the learned kernels are high-pass filters. This also eases the computational load by solely concentrating on the prediction of coefficients. Furthermore, we have explored the direct learning of the kernels without constraints. Our findings are shown in Table 4. This direct learning approach is equivalent to a "3D conv" or "Naive Kernels" (if we use linear combination). Our experimental results reveal a performance decline of 0.07dB and 0.09dB for these respective methods.
>
> *Q4: Comparisons with more recent methods.*
>
> **A4:** We mainly compare with recent methods of similar complexity in the paper. We will consider expanding our discussion to include additional recent methods in the revision.
>
> *Q5: The proposed method is not of "state-of-the-art performance".*
>
> **A5:** At this time, we don’t think a direct comparison to [b] and [c] based only on PSNR is reasonable nor grounds to reject our paper.  These two methods have significantly greater compute and runtime.  The FLOPs of ShiftNet [b] is 4.23 times greater than ours. RVRT [c] has at least 1.63 times the FLOPs of our approach. When considering runtime, ShiftNet is 9.09 times slower than our model, and RVRT lags behind by a factor of 3.15, largely due to its window and deformable attention mechanisms. Given these substantial disparities in complexity, drawing a fair comparison between these models proves challenging. It's worth noting that the reported FLOPs for ShiftNet [b] appear lower due to their use of a smaller input size in their calculations.
>
> *Q6: More comparisons on real examples.*
>
> **A6:** Due to the page limit, more comparisons are available in the supplementary material.

---

### Meta-Review · Area_Chair_SnEh · 2023-12-11

**Metareview:**

This paper aims to enhance the learning of latent high frequency information by merging the principles of the classic unsharp masking with a deep learning framework for better video deblurring.

It received reviews with mixed ratings. The major concerns include unclear motivation, insufficient experimental results, marginal performance improvement.

The rebuttal does not solve the concerns of reviewers well. Although the attitude of Reviewer 5xHX on this paper is positive, the review confidence rating is not high. Based on the recommendations of reviewers, the paper is not ready for ICLR.

**Justification For Why Not Higher Score:**

N/A

**Justification For Why Not Lower Score:**

N/A

---

### Decision · Program_Chairs · 2024-01-16

Reject